# Effects of PrObiotics on the Symptoms and Surgical ouTComes after Anterior REsection of Colon Cancer (POSTCARE): A Randomized, Double-Blind, Placebo-Controlled Trial

**DOI:** 10.3390/jcm9072181

**Published:** 2020-07-10

**Authors:** In Ja Park, Ju-Hoon Lee, Bong-Hyeon Kye, Heung-Kwon Oh, Yong Beom Cho, You-Tae Kim, Joo Yun Kim, Na Young Sung, Sung-Bum Kang, Jeong-Meen Seo, Jae-Hun Sim, Jung-Lyoul Lee, In Kyu Lee

**Affiliations:** 1Department of Colon and Rectal Surgery, Asan Medical Centre and University of Ulsan College of Medicine, 88 Olympic-ro 43-gil, Songpa-gu, Seoul 05505, Korea; ipark@amc.seoul.kr; 2Department of Food Science and Biotechnology, Graduate School of Biotechnology, Kyung Hee University, 1732, Deogyeong-daero, Giheung-gu, Yongin-si, Gyeonggi-do 17104, Korea; juhlee@khu.ac.kr (J.-H.L.); youeutae@khu.ac.kr (Y.-T.K.); 3Department of Surgery, St. Vincent’s Hospital, College of Medicine, The Catholic University of Korea, 93, Jungbu-daero, Paldal-gu, Suwon-si, Gyeonggi-do 16247, Korea; ggbong@catholic.ac.kr; 4Department of Surgery, Seoul National University Bundang Hospital, 300 Gumi-dong Bundang-gu, Seongnam-si, Gyeonggi-do 13620, Korea; crsohk@gmail.com (H.-K.O.); kangsb@snubh.org (S.-B.K.); 5Department of Surgery, Samsung Medical Center, Sungkyunkwan University School of Medicine, 81 Irwon-ro, Gangnam-gu, Seoul 06351, Korea; yongbeom.cho@samsung.com; 6R&BD Centre, Korea Yakult Co. Ltd., 577, Gangnam-daero, Seocho-gu, Seoul 05505, Korea; monera@re.yakult.co.kr (J.Y.K.); jhsim@re.yakult.co.kr (J.-H.S.); jlleesk@re.yakult.co.kr (J.-L.L.); 7National Cancer Control Institute, National Cancer Centre, 323 Ilsan-ro, Ilsandong-gu, Goyang-si Gyeonggi-do 10408, Korea; nayoungsung@hotmail.com; 8Division of Pediatric Surgery, Samsung Medical Centre, Sungkyunkwan University School of Medicine, 81 Ilwon-ro, Gangnam-gu, Seoul 06351, Korea; jm0815.seo@samsung.com; 9Department of Surgery, Division of Colorectal Surgery, Seoul St. Mary’s Hospital, College of Medicine, The Catholic University of Korea, 222 Banpo-daero, Seocho-gu, Seoul 06591, Korea

**Keywords:** gastrointestinal microbiome, quality of life, sigmoid neoplasms, anti-bacterial agents, treatment outcome

## Abstract

We investigated microbiota changes following surgical colon cancer resection and evaluate effects of probiotics on microbiota and surgical recovery. This randomized double-blind trial was performed at four medical centers in South Korea. Of 68 patients expected to undergo anterior sigmoid colon cancer resection, 60 were eligible, of whom 29 and 31 received probiotics and placebo, respectively, for four weeks, starting at one week preoperatively. Third- and/or fourth-week information on anterior resection syndrome (ARS), inflammatory markers, and quality of life was obtained. Stool sample analysis was conducted after randomization and bowel preparation and at three and four postoperative weeks. Bacteria were categorized into Set I (with probiotic effects) and II (colon cancer-associated). The probiotic group’s ARS score showed an improving trend (*p =* 0.063), particularly for flatus control (*p =* 0.030). Serum zonulin levels significantly decreased with probiotics. Probiotic ingestion resulted in compositional changes in gut microbiota; greater increases and decreases in Set I and II bacteria, respectively, occurred with probiotics. Compositional increase in Set I bacteria was associated with reduced white blood cells, neutrophils, neutrophil-lymphocyte ratio, and zonulin. *Bifidobacterium* composition was negatively correlated with zonulin levels in the probiotic group. Probiotics improved postoperative flatus control and modified postoperative changes in microbiota and inflammatory markers.

## 1. Introduction

Bowel dysfunction with symptoms similar to irritable bowel syndrome (IBS) or functional bowel disorder may occur after colon resection [1]. Probiotics are effective in alleviating postoperative symptoms [2,3]. Hence, probiotics may influence the extent of postoperative bowel dysfunction by altering the gut microbiota composition. However, there are insufficient studies on serial gut microbiome changes after surgically resecting colon cancer [4].

There is growing evidence of the importance of gut microbiota status for health. Microbiota are affected by diet, lifestyle, smoking, and living environments, as well as medical interventions (drug intake and surgery). Thus, interest in microbiota-based diagnosis or treatment of diseases has increased [5]. An association may exist between gut microbiota and various colorectal diseases, including colon cancer, inflammatory bowel disease (IBD), and IBS [2,3]. Surgery, mechanical bowel preparation, and antibiotics, which are all aspects of cancer treatment, are associated with compositional changes in gut microbiota [6,7]. Colon cancer is also associated with changes in gut microbiota composition. Observing changes in the microbiota before and after surgery in bowel cancer cases can thus be useful to explore their contribution to recovery.

We evaluated changes in microbiota composition and inflammatory markers associated with surgical resection for colon cancer and investigated the impact of probiotics on postoperative bowel dysfunction, microbiota composition, and inflammatory markers during surgical treatment for colon cancer.

## 2. Materials and Methods

### 2.1. Study Design

The PrObiotics on symptom and Surgical ouTCome after Anterior REsection of colon cancer (POSTCARE) study is a randomized, double-blind, multicenter, exploratory trial performed at four medical centers in South Korea. Patients were screened to determine their eligibility based on diagnosis with colorectal cancer (CRC). The study protocol was approved by the institutional review board of each participating institution (KC16HIME0819;2016-1181; B-1609/361-003; 2016-08-153).

### 2.2. Trial Registration

This trial was registered at www.clinicaltrials.gov (NCT03531606).

### 2.3. Patients

The inclusion criteria were age (18–75 years), histologically confirmed adenocarcinoma of the sigmoid colon, and indication for anterior resection. The exclusion criteria were use of antibiotics or probiotics in the week before eligibility screening, pre-existing fecal or urinary incontinence, metastatic CRC, preoperative endoscopic or symptomatic obstruction, regardless of decompression at the time of stoma formation or stent insertion, severe cerebrovascular or heart disease, pregnancy or lactation, IBD, history of chemotherapy or radiotherapy, symptoms of infectious or immunologic disease, abnormal creatinine (two-fold greater than the normal range), abnormal aspartate/alanine aminotransferase (three-fold greater than the normal range), uncontrolled hypertension or diabetes, and refusal to participate. All included patients provided written informed consent.

### 2.4. Randomization and Masking

Eligible participants (*n* = 68) were randomly assigned (1:1) to the probiotic group (*n* = 33) or placebo group (*n* = 35) group (Figure 1) using randomization numbers, with the sequence concealed from investigators. The randomization table, a sequence of random numbers (A and B) generated by the randomization program beginning with clinical trial subject number 1, was designed and reproduced using SAS^®^ (SAS Institute, Cary, NC, USA). The study process was centrally coordinated by a clinical research agent from NeoNutra Co., Ltd. (Seoul, Korea). Both patients and investigators were blinded to the assignments to the two groups.

### 2.5. Procedure

#### 2.5.1. Intervention and Surgical Resection

Participants received 2 g of placebo powder (350 mg of xylooligosaccharides and 36 mg of fructooligosaccharides as prebiotics without probiotic strains) or multi-strain probiotic powder twice daily for 4 weeks. Most placebo powder ingredients besides prebiotics were sugars (lactose, xylitol, and maltitol) and strawberry flavoring. Probiotic powder contained three probiotic strains (*Bifidobacterium animalis* subsp. *lactis* HY8002 (1 × 10^8^ cfu), *Lactobacillus casei* HY2782 (5 × 10^7^ cfu), and *Lactobacillus plantarum* HY7712 (5 × 10^7^ cfu)). Other ingredients were similar to placebo ingredients. The probiotic and placebo powder were identical in appearance and taste. After randomization, participants received either placebo or probiotic powder starting at one week before surgery and twice daily for 21 days after the day of surgery thereafter.

Preoperative mechanical bowel preparation was performed in all patients. On the day before surgery, 170 mL Picosolution (Pharmbio Korea, Inc., Seoul, Korea) was taken twice (5 pm and 8 pm); oral non-absorbable antibiotics (500 mg ciprofloxacin and 500 mg metronidazole) were prophylactically administered every 12 h after bowel preparation. All patients received prophylactic intravenous cephalosporin at 1 h before the operation (discontinued following a single postoperative administration) and underwent anterior resection.

#### 2.5.2. Fecal Sample Collection and Total Fecal DNA Extraction

During the trial, fresh fecal samples were collected as follows: (1) at baseline (visit 1, one week before surgery); (2) at one day before surgery (visit 2, after antibiotics and mechanical agent); (3) at three weeks after surgery (visit 6, at the completion of four-week test powder intake); and (4) at four weeks after surgery (visit 7, one week after discontinuing test powder intake) (Figure 2). Fecal samples were stored at −80 °C, and total fecal DNA extraction and purification followed previously optimized method [8].

#### 2.5.3. Bacterial Composition Analysis, Next-Generation Sequencing, and Bioinformatics Analysis

To understand the effects of surgery when antibiotics and probiotics were used, comparative analyses of bacterial composition were performed. The relative abundance of gut microbiota in the fecal samples from the placebo and probiotic groups was determined at the phylum and genus levels. Based on their general characteristics, bacteria were divided into two groups: Set I (*Bifidobacterium*, *Akkermansia*, *Parabacteroides*, *Veillonella*, *Lactobacillus*, and *Erysipelatoclostridium*) and Set II (*Prevotella*, *Alloprevotella*, *Fusobacterium*, and *Porphyromonas*). Set I bacteria had similar characteristics (fermentation activity with non-digestible complex carbohydrates and oligosaccharides) [9,10,11]. Set II bacteria, however, as suggested, are associated with intestinal diseases, including colon cancer [12]. For sequencing of the partial 16S rRNA, the V3/V4 region was PCR-amplified using extracted fecal DNA samples and the 341F (5′-CCTACGGGNGGCWGCAG-3′) and 805R_M (5′-GACTACHVGGGTMTCTAATCC-3′) universal primer set and was sequenced using an Illumina MiSeq sequencer according to the manufacturer’s standard protocol. For sequencing the complete 16S rRNA, PCR was amplified using the extracted fecal DNA samples and the 27F/1492R universal primer Set 1 and was sequenced using a PacBio RS II sequencer (Pacific Biosciences, Menlo Park, CA, USA). The qualified reads were assembled and statistically analyzed using the QIIME program. The compositional diversity of the gut microbiota was investigated with α- (Shannon index) and β-diversity analyses (redundancy analysis (RDA)), using QIIME, with the number of operational taxonomic units (OTUs).

### 2.6. Outcomes

The primary study outcome was improvement in anterior resection syndrome (ARS). This primary endpoint was assessed using an established ARS questionnaire [13]. The secondary outcomes were changes in fecal microbiota, inflammatory markers, nutritional screening index, quality of life (QoL), and postoperative complications. QoL was assessed twice (before and after the operation) using the European Organization for Research and Treatment of Cancer Quality-of-Life Questionnaire Core 30. To assess inflammatory markers, the Glasgow prognostic score (GPS), white blood cell (WBC) count, neutrophil count, lymphocyte count, monocyte count, platelet count, neutrophil-lymphocyte ratio (NLR), zonulin level, and cytokines (tumor necrosis factor-α (TNF-α), interferon-γ (IFN-γ), interleukin (IL)-6, and IL-10) were evaluated. Postoperative complications were categorized according to Clavien-Dindo classification. Changes in fecal microbiome were analyzed using the aforementioned method. The evaluation schedule of outcome variables is summarized in Figure 2.

Test powder administration was discontinued following any unexpected systemic disease, unacceptable adverse event (AE), or patient refusal. Physical and laboratory examinations, including chest and abdominopelvic computed tomography scans, were performed at two weeks before randomization. AEs were monitored during and after the study treatment and continued until the end of trial. Complete laboratory examinations, vital signs, body weight, and electrocardiography evaluations were performed. All AEs reported during the study period were coded according to the Medical Dictionary for Regulatory Activities version 21.0. AEs occurring after the test powder administration were tabulated, and their incidence was calculated and evaluated. Use of any other probiotics was prohibited during the trial to prevent possible interference with safety, tolerability, or efficacy evaluations.

### 2.7. Statistical Analysis

This study, comprising exploratory trials to assess the clinical efficacy of probiotics, was designed to recruit 30 patients per group at a significance level of 5%, power of 80%, and minimum effect size of 0.735. Safety and complications were analyzed in the safety set, whereas other variables were analyzed in the full analysis set (Figure 1). Baseline demographics of the two groups were compared using the chi-square or Fisher’s exact test for qualitative variables and Student’s t-test for quantitative variables. Primary and secondary outcomes were compared via a simple comparison between the two groups and through serial changes from the baseline visit. For simple comparisons, the Wilcoxon rank-sum test was used, and serial changes were analyzed using generalized estimating equations.

Any *p* < 0.05 after adjustment for multiple comparisons was considered significant. All statistical analyses were performed using SAS version 9.4 (SAS Institute, Cary, NC, USA).

All co-authors have access to the study data and have reviewed and approved the final manuscript.

### 2.8. Patient and Public Involvement

There was no patient and public involvement in the study design, recruitment, and conduct of this study.

## 3. Results

### 3.1. Patient Characteristics and Postoperative Complications

Of 68 recruited patients who were to undergo anterior resection for sigmoid colon cancer, 60 were eligible (29 and 31 in the probiotic and placebo groups, respectively). No significant differences in baseline and pathological characteristics were observed between the two groups (Table 1).

AEs were evaluated in the safety set, and these are not different between the groups (*p =* 0.297) Among the AEs that developed in 11 patients (33.3%) in the probiotic group, none were severe, and hepatobiliary disorder was the most common. Among 22 AEs in 16 patients (45.7%) in the placebo group, two (one anastomotic leakage, one intractable diarrhea) were severe, and the patients were excluded out from the study. Gastrointestinal discomfort was the most common AE in the placebo group (*n* = 7, 31.8%) (Appendix A). Postoperative complications (Clavien-Dindo classification ≥2) occurred in 2 (6.0%) and 10 (28.5%) patients in the probiotic and placebo groups, respectively (*p =* 0.024). In the probiotic group, cholecystitis and postoperative pneumonia occurred. In the placebo group, surgical complications developed in three patients (wound infection, ileus, and anastomotic leakage in one patient each), and the remaining seven patients had nonsurgical complications, including fever (two patients), abnormal liver function test (one patient), enteritis (one patient), cystitis (one patient), pneumothorax (one patient), and pseudomembranous colitis (one patient). Non-surgical infectious complications occurred in two and five patients in the probiotic and placebo groups, respectively.

### 3.2. Changes in Anterior Resection Syndrome

At baseline, the placebo group showed significantly higher ARS score (*p =* 0.016), although no difference in major ARS proportions was noted (*p =* 0.282). No difference in ARS scores was observed between groups after visit 6, after completing probiotic or placebo powder administration. In both groups, the ARS score significantly changed before and after operation (*p =* 0.018), with less prominent changes in the probiotic group from baseline to visit 6 (*p =* 0.053; Figure 2). Despite the increased proportion of patients with major ARS in both groups from baseline to final evaluation (3.2% vs. 16.7% for placebo and 10.3% vs. 17.2% for probiotic), the changes within groups were not significant. Flatus control significantly improved in the probiotic group. Despite a steady decrease in flatus control at visits 6 (67.7%) and 7 (56.7%) in the placebo group, it increased to 72.4% and 69%, respectively, in the probiotic group. Those who showed improvement were significantly higher in the probiotic group (31%) than in the placebo group (10%) (*p =* 0.033) (Figure 2 and Appendix A).

No difference in QoL and Nutritional Screening Index was noted between the groups (on any of the functional or symptom scales).

### 3.3. Changes in Inflammatory Markers

Among inflammatory markers, only zonulin showed differences between groups at baseline (*p =* 0.027), which was eliminated at visit 6. Moreover, the zonulin level decline from baseline to visit 6 was three times greater in the probiotic group than in the placebo group (66% vs. 22%) (*p =* 0.035; Figure 2 and Appendix A). The lymphocyte and platelet counts increased at visit 6 compared to baseline, whereas IFN-γ, TNF-α, IL-10, and IL-6 showed no noticeable change in either group. Neutrophils, GPS, and NLR increased until Visit 5 and subsequently decreased at Visit 6 (Figure 2 and Appendix A).

### 3.4. Changes in Microbiota Composition

Overall, 236 fecal samples from 59 patients (four per patient) were collected at four time points (Figure 2). Total fecal DNA was successfully extracted from 228 fecal samples for metagenomic analysis. Random sequencing of 16S rRNA amplicons from the extracted fecal DNA yielded 277,552 reads and 873 OTUs (915 and 830 for the placebo and probiotic groups, respectively) (Appendix A). The α-diversity analysis showed significant reduction in the Shannon index for fecal samples taken between Visits 2 and 6 in both groups (Figure 3a). Although the Shannon index at visits 6 and 7 did not differ between the groups, while the average Shannon index at visit 7 slightly increased in the placebo group, it slightly decreased in the probiotic group. Although some samples in these four groups overlapped in the RDA graph, they were positioned differently (Figure 3b,c). The pre- and postoperative samples of patients in the placebo group were scattered in different positions, while the placebo and probiotic groups’ positions after surgery indicated differences in β-diversity results in the RDA graph (Figure 3d).

Over 98% of bacteria in the samples belonged to five dominant bacterial phyla (*Firmicutes*, *Bacteroidetes*, *Actinobacteria*, *Proteobacteria*, and *Verrucomicrobia*) and were categorized into Set I or II. Comparative analysis of gut microbiota compositions between visits 2 and 3, following bowel preparation and at 1 week of probiotic intake, showed no significant compositional changes (Appendix A). All selected major genera in Set I were significantly increased by >1000% in both groups, whereas other selected major genera in Set II decreased between visits 3 and 6 except *Fusobacterium* (Table 2 and Appendix A). The relative abundance of Set II bacteria *Alloprevotella* and *Porphyromonas* drastically decreased at visit 6 in the probiotic group when compared to their increased composition rates observed at visit 3. Even after discontinuing test powder intake, the relative abundance for both Set I and Set II remained unchanged and was maintained between visits 6 and 7 (Table 2). The relative abundance of *Alloprevotella* and *Porphyromonas* in Set II remained extremely low at visit 7 in the probiotic group.

Although increased compositional ratios of the selected major bacteria in Set I occurred in both placebo and probiotic groups, the Set I bacteria ratios were much higher in the probiotic group, whereas the Set II bacteria ratios were lower in the probiotic group than in the placebo group. After completing the test powder intake, the outcome measurements differed from the baseline evaluation in the two groups (Table 2).

It was difficult to clarify the correlations between changes in gut microbiota composition and biomarkers in the placebo group. However, the patterns of correlation between Set I/Set II bacteria in the probiotic group and the biomarkers were clearly different. In the probiotic group, Set I bacteria were accompanied by decreases in specific biomarkers, including WBC, neutrophils, NLR, and zonulin, while Set II bacteria were associated with increased IL-10 and IFN-γ. In particular, a negative correlation between *Bifidobacterium* composition and zonulin level was found in the probiotic group (Figure 4).

The changes in metabolite composition between groups were also investigated. Analysis of the correlation between the major gut bacteria (52 genera) and 75 different metabolites revealed some correlations between specific gut bacteria and specific metabolites only in the probiotic group (Appendix A). Additional correlation analyses between the gut Set I/Set II bacteria and those metabolites, showed that this correlation occurred only in the probiotic group (Appendix A).

## 4. Discussion

This study showed that changes in postoperative bowel dysfunction were slow with the use of probiotics. Inflammatory marker changes differed by probiotic administration due to the beneficial microbiota alteration. Intake of probiotics resulted in an increase of short-chain fatty acid-producing bacteria, a greater decrease in bacteria associated with colon cancer development, and improved postoperative recovery (in inflammatory change and defecatory dysfunction).

Complications directly related to anterior resection occurred only in the placebo group and included one anastomotic leakage case. Although no difference existed in surgical complications (≥ Clavien-Dindo Grade ≥ 2), between the two groups, the low incidence might have resulted from insufficient statistical power, necessitating a larger sample size in future studies. In addition to surgical complications, infectious complications are important. Probiotics are widely used; however, their efficacy and safety are controversial. Septicemia has been reported in immunocompromised patients or in those with medical co-morbidities [14,15]. However, such cases were resolved with surgery, antibiotic treatment, or conservative management. Effects and AEs of probiotic administration before and after gastrointestinal surgery are insufficiently studied. A few studies showed that probiotic administration increased infection and mortality [16,17]. In the present study, however, life-threatening infectious complications did not occur. We excluded patients with immunocompromised status and severe medical co-morbidities, which reflected that probiotics were safe in patients with colorectal cancer without detrimental medical conditions.

In this study, the incidence rates of AEs due to test and placebo powders did not significantly differ. Among the reported AEs, those with a clear association with test powder were reported. However, we could not exclude the relationship of the two AEs with the test powder in the probiotics group (cholelithiasis and tremor) and three AEs in the placebo group (diarrhea, tremor, and abnormal liver function). Patients with AEs potentially related to test powder recovered without specific treatment after discontinuation of test powder on completion of the study and were included in the final analysis. However, further studies are required to determine the relationship between the reported AEs and test powder.

Bowel dysfunction after low anterior resection has multifactorial causes that can lead to QoL decline [18,19]. A few reports have described bowel dysfunction after sigmoid colon resection in diverticular disease [20,21], including a recent Danish population-based cross-sectional study [21,22]. Probiotic intake was also associated with decreased ARS, which increased at one week after probiotic discontinuation, indicating that treatment discontinuation may prevent the beneficial effect on ARS. Long-term bowel dysfunction should be serially evaluated to determine its natural course and the effect of probiotics.

Only zonulin, a clinical measure of bacterial translocation, was significantly different between groups. Zonulin can regulate tight junctions and is implicated in regulating mucosal permeability [23,24]. Increased zonulin is associated with increased mucosal permeability. In this study, zonulin significantly decreased in the probiotic group compared with the placebo group. This finding indicates the muco-protective effect of probiotics on zonulin levels to reduce mucosal permeability. Dysregulation of the zonulin pathway was reported in IBS and IBD [25,26]. However, association between postoperative bowel dysfunction and zonulin is not well studied.

Although not statistically significant, the pattern of change in the gut microbiota differed between the two groups. There was a larger decrease in NLR and lymphocytes in the probiotic group than in the placebo group, indicating that the levels of acute inflammation/infection-related markers were lower in the probiotic group. Cytokines related to inflammation also showed different trends. INF-γ and IL-10, which are anti-tumor and anti-inflammatory cytokines, were increased in the probiotic group but were decreased or unchanged in the placebo group. Albeit not statistically significant, this inverse trend implied that probiotics modified the immune and inflammatory responses to surgery. The associations between gut microbiota and immune response are well studied [27,28], and the Human Functional Genomics Project showed associations between gut microbiota and stimulus-induced cytokine responses [27]. Gut microbial species and genera were significantly associated with inflammatory cytokine responses.

The probiotics used in this study have been used in fermented milk for a long time and are proven to be safe by various intake histories. Before this clinical study, we conducted an in vitro experiment to determine the immunomodulatory effect of probiotics comprising *L. plantarum* HY7712, *L. casei* HY2782, and *B. lactis* HY8002. Dendritic cells isolated from mice were treated with each probiotic strain and co-cultured with T-cells, also isolated from mouse spleens. All three probiotic strains increased the number of IFN-γ-secreting T-cells but did not affect TNF-α secretion. *B. lactis* HY8002 and *L. plantarum* HY7712 increased IL-10 secretion, an anti-inflammatory cytokine, and increased Foxp3 presented CD4 T cells (Tregs). *L. casei* HY2782 did not affect IL-10 expression or the number of Tregs. Interestingly, *L. casei* HY2782 greatly increased the expression of IL-12 in dendritic cells, while the other two probiotics showed less response in dendritic cells.

Altogether, these findings suggest that the probiotic strain used in this study improved the Th1 response associated with enhanced immunity and activated Treg cells that regulate hypersensitive immunity. Therefore, the probiotics combination used in this study can effectively improve the immune system balance.

Statistical analysis of compositional diversity in the gut microbiota of patients in the placebo and probiotic groups revealed the impacts of surgery in conjunction with the use of antibiotics and probiotics. Diversity analysis showed a slight reduction in the Shannon index only in the probiotic group after discontinuing probiotic treatment, probably due to the emergence of dominant genera from specific probiotics. The effects of colon cancer surgery and probiotic ingestion on compositional changes have been previously reported [29].

The use of oral antibiotic preparation, a well-established bowel preparation method, in addition to MBP for colon surgery affects the probiotics and gut microbiota. The type of antibiotics could be associated with the extent of change in the microbiota [30]. In the present study, we used oral metronidazole and ciprofloxacin as antibiotic bowel preparation agents because we wanted to cover infections caused by both aerobic and anaerobic bacteria. Although recent studies reported that oral non-absorbable antibiotics such as kanamycin or neomycin with metronidazole were effective in decreasing surgical site infection [31,32], ciprofloxacin was also used as an oral antibiotic agent because of its high bioavailability and good tolerance. In addition, ciprofloxacin is readily available in this region. We hypothesized that the combination of two absorbable antibiotics might eliminate the gut microbiota as well as probiotics and that surgery might influence the composition of the gut microbiota.

To extend our understanding of these impacts, the compositional changes in gut microbiota at each sampling time point were compared. Before surgery, initial probiotic ingestion helped prevent Set I bacteria reduction in the probiotic group. However, the initial probiotic ingestion period was too short to draw conclusions regarding the role of probiotics in compositional change. The relative compositional abundance of all genera was maintained despite washout using MBP and bowel preparation using oral antibiotics, indicating that the effect of previous probiotics may remain active.

Subsequent comparisons of compositional changes with long-term probiotics use revealed no significant difference between the placebo and probiotic groups, indicating that probiotic ingestion may not be effective for enhancing beneficial Set I bacteria composition. Nonetheless, *Alloprevotella* and *Porphyromonas* were almost completely wiped out in the probiotic group only, suggesting the importance of probiotics for compositional changes in the gut environment. Probiotics may be associated with recovery of gut microbiota after colon cancer surgery, even during this washout period.

Effective stimulation of beneficial bacteria growth in Set I and inhibition in Set II by probiotics may be closely associated with the effective recovery of damaged gut microbiota in patients after colon cancer surgery. Furthermore, ingesting only prebiotic powder in the placebo group showed similar, but weaker, effect on compositional regulation of gut microbiota, suggesting that ingesting a probiotic and prebiotic combination is more effective than probiotics consumption alone. Metabolome analysis of human fecal samples revealed that some gut bacteria, including *Prevotella*, *Blautia*, *Alloprevotella*, *Ruminococcus*, *Faecalibacterium*, and *Butyricicoccus*, had positive correlations with specific metabolic acids from peptides, carbohydrates, and lipids only in the probiotic group. While their positive correlations are not yet completely understood, compositional changes in the gut microbiota of patients caused by CRC surgery and probiotic ingestion likely led to partial metabolomic changes.

This study has some limitations. We analyzed changes in microbiota composition and clinical, inflammatory, and immunologic markers; nonetheless, direct evidence of the association between microbiota changes and clinical, inflammatory, and immunologic marker changes cannot be inferred from our findings. Another limitation was the lack of long-term follow-up. After probiotic discontinuation, the remedying effects did not persist; however, patients were only followed up for one week after probiotic discontinuation. Finally, prebiotics were used in the placebo group. The study group received both prebiotics and probiotics, and the probiotic effect was assumed to have occurred during post-surgical recovery from colon cancer surgery; nevertheless, the natural course of these patients in terms of fecal microbiota and inflammatory and immunologic marker changes without any intervention was not evaluated.

## 5. Conclusions

In summary, postoperative microbiomic changes typically occur over time in CRC patients. Pre- and postoperative treatments and surgery for patients with colon cancer result in changes in the microbiota and inflammatory responses. These changes in the microbiota and inflammatory responses are modified by the use of probiotics before and after surgery, which reduces postoperative bowel discomfort, as expressed by the anterior resection syndrome score. These results suggest that probiotics are useful in maintaining homeostasis when intestinal microbiota changes are caused by any inflammatory reactions in normal subjects. This possibility warrants further research.

## Figures and Tables

**Figure 1 jcm-09-02181-f001:**
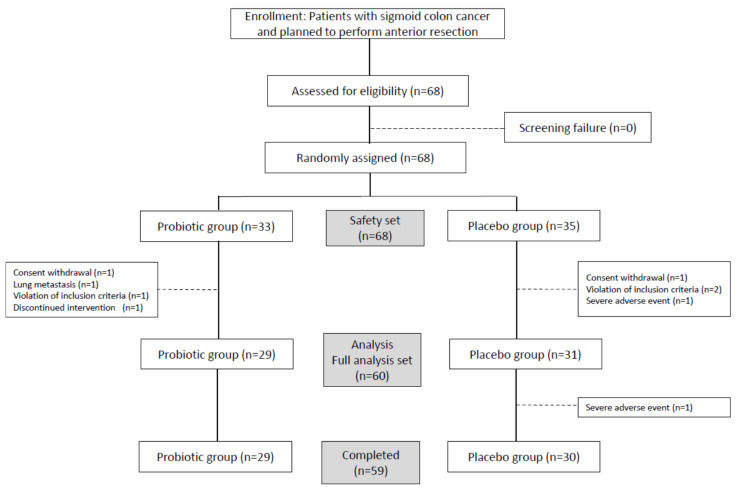
Study design. Consort diagram.

**Figure 2 jcm-09-02181-f002:**
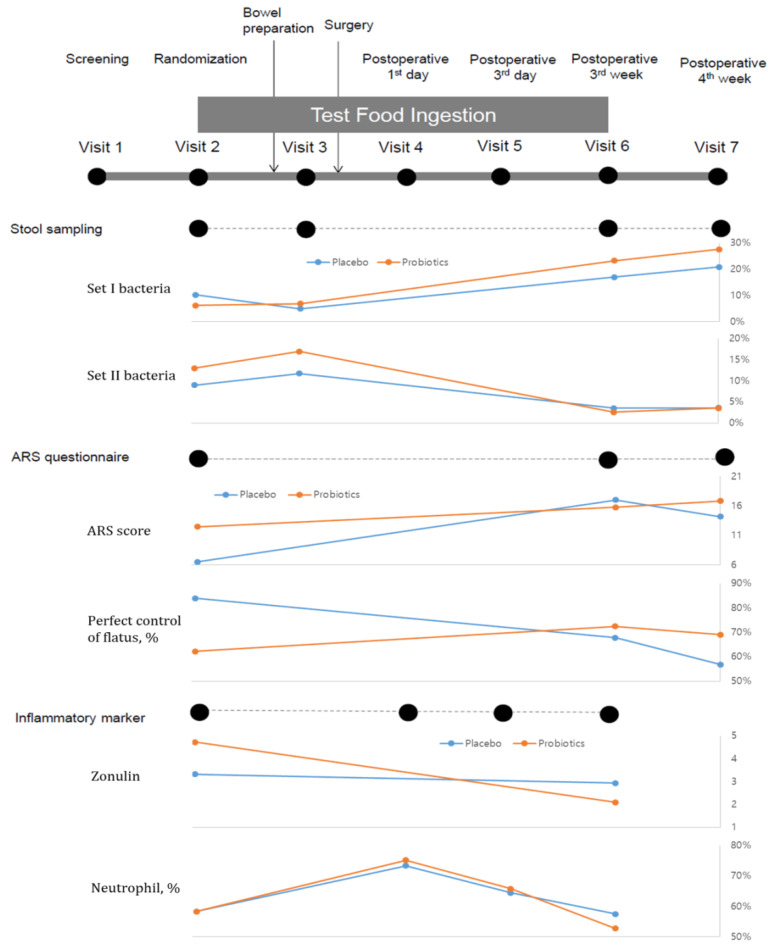
Clinical and laboratory analysis and fecal sampling schedule during the study period. ARS, anterior resection syndrome.

**Figure 3 jcm-09-02181-f003:**
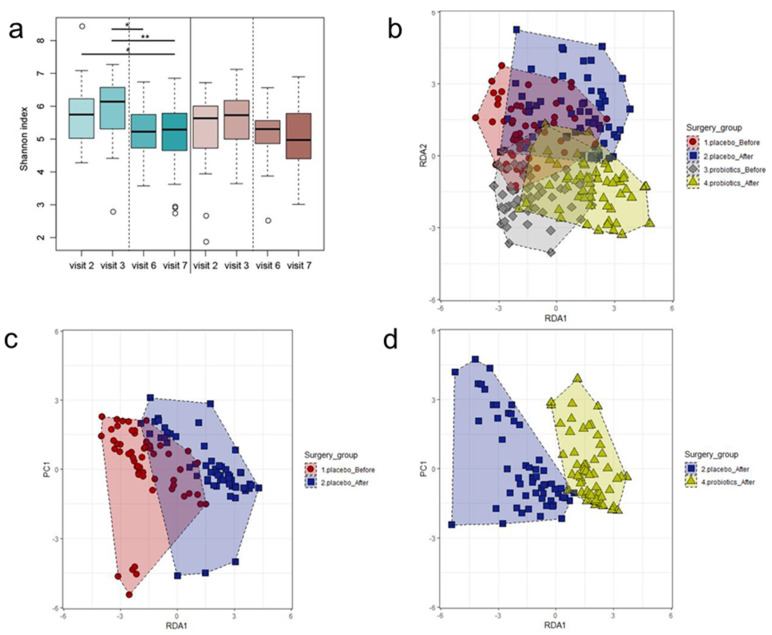
Diversity analysis at each time point. (**a**) α-Diversity analysis between each visit’s fecal sample for both groups. (**b**) β-Diversity of the placebo and probiotics group categorized as before and after surgery. (**c**) β-Diversity of the placebo group before and after surgery. (**d**) β-Diversity before and after surgery between the placebo and probiotics groups. RDA, redundancy analysis.

**Figure 4 jcm-09-02181-f004:**
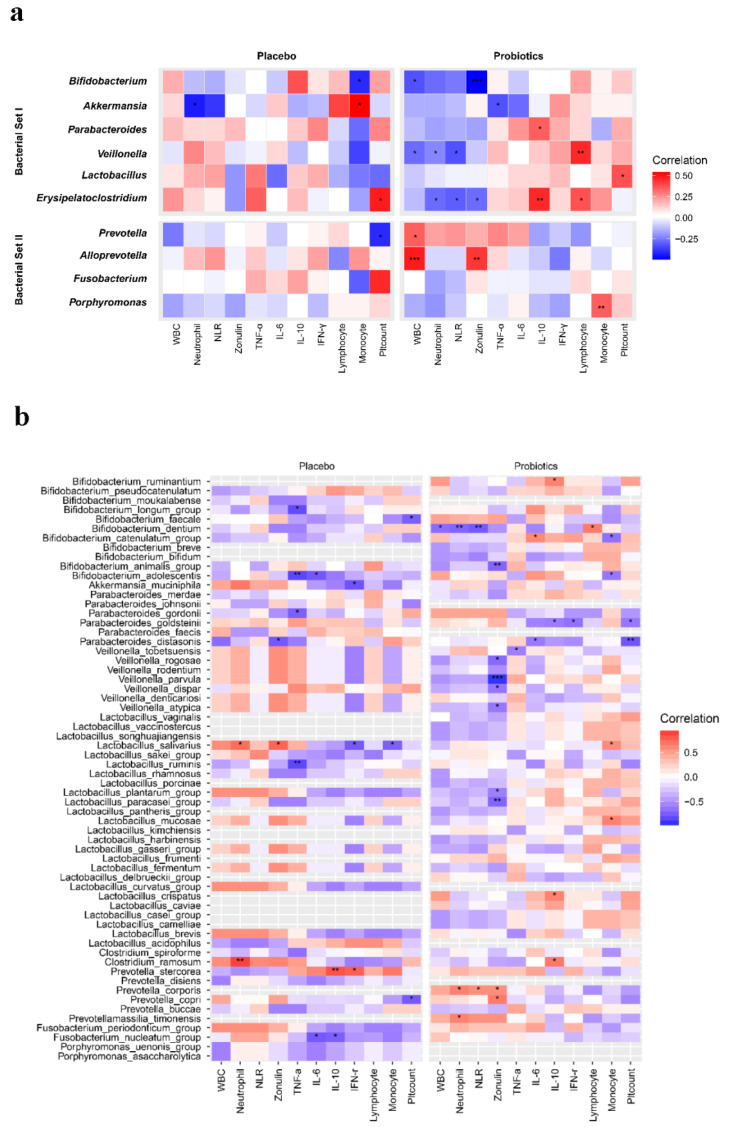
Spearman correlation analysis between clinical data and bacterial sets for the probiotics and placebo groups. (**a**) Bacterial Set I negatively correlated to WBC, neutrophils, NLR, and zonulin. Conversely, Bacterial Set II showed a reverse correlation. No correlation between biomarkers and bacterial sets composition showed in the placebo group. (**b**) Spearman correlation analysis in bacterial species level with biomarkers. WBC, white blood cells; NLR, neutrophil-lymphocyte ratio; IFN, interferon; TNF, tumor necrosis factor; IL, interleukin.

**Table 1 jcm-09-02181-t001:** Patient demographic and pathology data.

		Placebo (*n* = 31)	Probiotics (*n* = 29)	*p*-Value
Age, years, mean (SD)		61.03 (7.02)	60.10 (10.37)	0.69 *
Sex	Male	13 (41.94)	19 (65.52)	0.07 ^†^
	Female	18 (58.06)	10 (34.48)	
BMI, kg/m^2^		24.04 (3.53)	24.42 (2.90)	0.65 *
Exercise	No	15 (48.39)	20 (68.97)	0.31 ^‡^
Times/week	<3	5 (16.13)	3 (10.34)	
	≥3	11 (35.48)	6 (20.69)	
Alcohol	No	16 (51.61)	14 (48.28)	0.56 ^‡^
Bottles/week	Former use	2 (6.45)	6 (20.69)	
	<1	4 (12.90)	3 (10.34)	
	<4	3 (9.68)	3 (10.34)	
	≥4	6 (19.35)	3 (10.34)	
Smoking	None	16 (51.61)	14 (48.28)	0.74 ^†^
	Ex-smoker	7 (22.58)	9 (31.03)	
	Smoker	8 (25.81)	6 (20.69)	
Comorbidities		16 (57.1)	21 (71.4)	0.227 ^†^
Pathological stage	Stage I	6 (19.4)	12 (41.4)	0.148
	Stage II	8 (25.8)	6 (20.7)	
	Stage III	15 (48.4)	11 (42.3)	
	Stage IV	2 (6.5)	0 (0)	
No. of harvested LNs, mean (SD)		24.6 (12.6)	22.6 (7.8)	0.469 *
Lymphatic invasion	Negative	19 (61.3)	22 (75.9)	0.23 ^†^
	Positive	12 (38.7)	7 (24.1)	
Vascular invasion	Negative	22 (71.0)	25 (86.2)	0.15 ^†^
	Positive	9 (29.0)	4 (13.8)	
Perineural invasion	Negative	19 (61.3)	24 (82.8)	0.65 ^†^
	Positive	12 (38.7)	5 (17.2)	

*: *p*-value obtained using two-sample t-test. ^†^: *p*-value obtained using chi-square test. ^‡^: *p*-value obtained using Fisher’s exact test. SD, standard deviation; BMI, body mass index; LN, lymph node.

**Table 2 jcm-09-02181-t002:** Compositional change in the bacterial set between sampling time points.

**Visit 2 vs. 3**	**Genus**	**Placebo-Visit 2**	**Placebo-Visit 3**	**Ratio**	**Probiotics-Visit 2**	**Probiotics-Visit 3**	**Ratio**
Set I bacteria	Bifidobacterium	4.388%	2.810%	−36.0%	3.638%	3.225%	−11.3%
	Akkermansia	0.859%	0.229%	−73.3%	1.178%	1.338%	13.5%
	Parabacteroides	1.914%	0.561%	−70.7%	0.397%	0.507%	27.7%
	Veillonella	0.448%	0.176%	−60.8%	0.226%	1.192%	428.3%
	Lactobacillus	2.231%	1.084%	−51.4%	0.669%	0.526%	−21.3%
	Erysipelatoclostridium	0.419%	0.085%	−79.7%	0.077%	0.072%	−6.0%
Set II bacteria	Prevotella	7.368%	9.650%	31.0%	10.286%	15.469%	50.4%
	Alloprevotella	0.687%	0.822%	19.6%	1.053%	1.368%	29.9%
	Fusobacterium	0.174%	0.215%	23.3%	1.176%	0.041%	−96.6%
	Porphyromonas	0.713%	1.050%	47.3%	0.465%	0.012%	−97.4%
**Visit 3 vs. 6**	**Genus**	**Placebo-Visit 3**	**Placebo-Visit 6**	**Ratio**	**Probiotics-Visit 3**	**Probiotics-Visit 6**	**Ratio**
Set I bacteria	Bifidobacterium	2.810%	6.873%	* 144.6%	3.225%	9.484%	** 194.1%
	Akkermansia	0.229%	2.224%	870.1%	1.338%	4.164%	211.2%
	Parabacteroides	0.561%	3.715%	** 562.6%	0.507%	5.836%	* 1051.3%
	Veillonella	0.176%	2.445%	* 1291.6%	1.192%	2.483%	108.2%
	Lactobacillus	1.084%	1.195%	10.2%	0.526%	0.875%	66.2%
	Erysipelatoclostridium	0.085%	0.541%	* 535.7%	0.072%	0.356%	* 393.4%
Set II bacteria	Prevotella	9.650%	2.214%	* −77.1%	15.469%	2.292%	*** −85.2%
	Alloprevotella	0.822%	0.097%	−88.2%	1.368%	0.004%	** −99.7%
	Fusobacterium	0.215%	0.917%	326.3%	0.041%	0.227%	459.2%
	Porphyromonas	1.050%	0.287%	−72.7%	0.012%	0.001%	* −91.0%
**Visit 6 vs. 7**	**Genus**	**Placebo-Visit 6**	**Placebo-Visit 7**	**Ratio**	**Probiotics-Visit 6**	**Probiotics-Visit 7**	**Ratio**
Set I bacteria	Bifidobacterium	6.873%	7.339%	6.8%	9.484%	9.961%	5.0%
	Akkermansia	2.224%	7.041%	216.7%	4.164%	9.256%	122.3%
	Parabacteroides	3.715%	4.203%	13.1%	5.836%	3.847%	−34.1%
	Veillonella	2.445%	0.660%	−73.0%	2.483%	2.786%	12.2%
	Lactobacillus	1.195%	0.917%	−23.3%	0.875%	1.239%	41.6%
	Erysipelatoclostridium	0.541%	0.666%	23.2%	0.356%	0.487%	37.1%
Set II bacteria	Prevotella	2.214%	2.562%	15.7%	2.292%	3.287%	43.4%
	Alloprevotella	0.097%	0.091%	−5.9%	0.004%	0.004%	−7.9%
	Fusobacterium	0.917%	0.620%	−32.4%	0.227%	0.237%	4.4%
	Porphyromonas	0.287%	0.302%	5.3%	0.001%	0.000%	−79.3%
**Visit 2 vs. 7**	**Genus**	**Placebo-Visit 2**	**Placebo-Visit 7**	**Ratio**	**Probiotics-Visit 2**	**Probiotics-Visit 7**	**Ratio**
Set I bacteria	Bifidobacterium	4.388%	7.339%	67.2%	3.638%	9.961%	* 173.8%
	Akkermansia	0.859%	7.041%	720.1%	1.178%	9.256%	* 685.4%
	Parabacteroides	1.914%	4.203%	119.6%	0.397%	3.847%	* 869.3%
	Veillonella	0.448%	0.660%	47.4%	0.226%	2.786%	* 1134.4%
	Lactobacillus	2.231%	0.917%	−58.9%	0.669%	1.239%	85.1%
	Erysipelatoclostridium	0.419%	0.666%	58.9%	0.077%	0.487%	535.9%
Set II bacteria	Prevotella	7.368%	2.562%	−65.2%	10.286%	3.287%	* −68.0%
	Alloprevotella	0.687%	0.091%	−86.7%	1.053%	0.004%	* −99.7%
	Fusobacterium	0.174%	0.620%	255.5%	1.176%	0.237%	−79.9%
	Porphyromonas	0.713%	0.302%	−57.6%	0.465%	0.000%	−100.0%

* *p*-value <0.05; ** *p*-value; *** *p*-value <0.001.

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
