# Peer review of "Effects of PrObiotics on the Symptoms and Surgical ouTComes after Anterior REsection of Colon Cancer (POSTCARE): A Randomized, Double-Blind, Placebo-Controlled Trial"

_jcm, 2020, doi:10.3390/jcm9072181_

Round 1
Reviewer 1 Report
This article is well-written and treats a very interesting problem of microbiota modulations. Despite the limitations, this study does provide new prospects in the role of probiotics on the recovery of gut microbiota after colon cancer surgery the management of colorectal cancer. Longer follow up period should be considered as well as the prolonged probiotic intervention (6-8 weeks). It would be worth adding the placebo group without any interventions.
Author Response
We greatly appreciate your thoughtful comments and completely agree with your opinion. We need longer follow-up to evaluate the long-term effects of probiotics with prolonged intervention. Comparison of prolonged administration of probiotic with a time-limit would also be required, as mentioned by the reviewer. The use of prebiotics in the control group is a limitation to analyzing the true changes in microbiota after colon cancer surgery. However, we consider that this study could suggest the role of probiotics on postoperative recovery and provide evidence for further study.
We are planning to conduct further study to evaluate the effects of probiotics and prebiotics on postoperative recovery and study the influence of the duration of intervention and possible mechanism involved in the process. We expect to find answers to these questions in future studies.
Reviewer 2 Report
Thank you to give me the chance to review this interesting article. Here are my comments.
Line 109: oral non-absorbable antibiotics (500 mg ciprofloxacin and 500 mg metronidazole).
Ciprofloxacin is almost entirely absorbed in the gastrointestinal tract as well as metronidazole, with only 6-15% of drug occurring in the colon after administration. Hence, neither ciprofloxacin nor metronidazole are oral non-absorbable antibiotics. Please specify why you choose ciprofloxacin in your current practice despite the Literature suggests the use of non-absorbable drugs like neomycin with either erythromycin (which is not worldwide approved) or metronidazole.
Moreover, do you think it is reasonable to give both an absorbable antibiotic and probiotics? Or do the antibiotics clear the probiotic?
There might be some confounding factors that are not addressed in the paper and that should be specified. Potential risk factors for Anterior Resection syndrome are: level of anastomosis (low anastomosis have higher risk), total as opposed to partial mesorectal excision, obstructive presenting symptoms. As a matter of fact you don’t consider obstructive presenting symptoms as an exclusion criteria: does it mean no one among your patients had colonic obstruction? Please address all cofounding factors.
Table 1: p-value for comorbidities is not specified. Please address.
The adverse events and complication paragraph should be rephrased. As it is not summarized in a table it is difficult to follow in the text. Line 189 you start with “of AEs occurring in 11 patients…” and in the following sentence “of 22 AEs occurring in the placebo group…”. So, you can’t compare how many patients with adverse events unless you mean that 11 patients in the probiotic group had adverse events and 22 patients in the placebo group. The p-value mentioned of 0.297 refers to the difference between AEs in the placebo vs probiotic group? 22 vs 11? No significance?
Which are the severe adverse events in the placebo group? And how do you explain them?
As far as complications is concerned, I would specify the infections developed in the two groups as there is concern that, in immunocompromised patients, probiotics can favor septic infections.
In the discussion you mention that adverse events of probiotic administration and placebo did not differ between the two groups, and all adverse events completely resolved by discontinuing the probiotic powder (Line 287-290). In the results chapter you never mention probiotic/placebo discontinuation. If so many patients discontinued the probiotic I don't think this drop out rate can support the conclusions you make.
In the discussion you should address the fact that the placebo group had more postoperative complications and more adverse events and possibly address it.
The study is well conducted and results are interesting but I do think that we should address all the confounding factors.
Author Response
Thank you to give me the chance to review this interesting article. Here are my comments.
Line 109: oral non-absorbable antibiotics (500 mg ciprofloxacin and 500 mg metronidazole).
Ciprofloxacin is almost entirely absorbed in the gastrointestinal tract as well as metronidazole, with only 6-15% of drug occurring in the colon after administration. Hence, neither ciprofloxacin nor metronidazole are oral non-absorbable antibiotics. Please specify why you choose ciprofloxacin in your current practice despite the Literature suggests the use of non-absorbable drugs like neomycin with either erythromycin (which is not worldwide approved) or metronidazole.
Moreover, do you think it is reasonable to give both an absorbable antibiotic and probiotics? Or do the antibiotics clear the probiotic?
(Answer) We appreciate your insightful comment and apologize for erroneously writing “non-absorbable” antibiotics. We have revised it as “absorbable antibiotics”.
As you pointed out, previous studies have reported the effects of combination of non-absorbable antibiotics, such as kanamycin or neomycin and metronidazole on reducing surgical-site infections < ST McSorley, et al. Meta-analysis of oral antibiotics, in combination with preoperative intravenous antibiotics and mechanical bowel preparation the day before surgery, compared with intravenous antibiotics and mechanical bowel preparation alone to reduce surgical-site infections in elective colorectal surgery. BJS Open, 2018;2 : 185-194; Elaine Vo, et al. Association of the Addition of Oral Antibiotics to Mechanical Bowel Preparation for Left Colon and Rectal Cancer Resections With Reduction of Surgical Site Infections. JAMA surg 2018; 153:114-121>.
However, no concrete recommendations have been provided for the administration of oral antibiotics before colorectal surgery. In our study, we used a combined oral regimen of ciprofloxacin and metronidazole. We tried to cover infections caused by both aerobic and anaerobic bacteria using this combination. Metronidazole has been well established as a preoperative prophylactic agent for colon cancer surgery. We chose ciprofloxacin because it has a high bioavailability and excellent tolerance <RL NicholsThe role of quinolones in abdominal surgerySurg Infect (Larchmt), 200;1: 65-72>. In addition, it can maintain good tissue diffusion at the time of surgery and is excreted mainly via hepatic metabolism; therefore, it would reduce the fecal bacterial load. It also has an effect on Pseudomonans aeruginosa, which negatively affects anastomotic healing. Although we must consider the risk of antibiotic resistance and increased risk of Clostridioides difficile with extended use of ciprofloxacin, 1-day preoperative prophylaxis is unlikely to result in these complications In the present study, none of the patients had C. difficile-associated colitis.
Therefore, we decided that the combination of metronidazole and ciprofloxacin would be beneficial to reduce surgical-site infections; in addition, both antibiotics are easy to deliver and readily available in this region.
As mentioned by the reviewer, the gut microbiota as well as probiotics may be damaged by the antibiotics < Panda S, et al. Short-term effect of antibiotics on human gut microbiota. PLoS One 2014;9:e95476; Willmann M, et al. Distinct impact of antibiotics on the gut microbiome and resistome: a longitudinal multicenter cohort study. BMC Biol 2019;17:76>
Preoperative oral antibiotic preparation is a well-established method to reduce SSI in colon cancer surgery, and it is a practical step for colon cancer surgery. The effect of antibiotics on the gut microbiota and probiotics is a cause for concern. We would like to know whether preparation with probiotics for some duration would alleviate the changes in gut microbiota by antibiotics preparation. We evaluated the bacterial composition in feces between Visit 2 (at randomization) and Visit 3 (after bowel preparation). The decrease in set I bacteria was lower in the probiotics group than in the placebo group; however, the effect on set II bacteria was heterogeneous. We believe that probiotics might have prevented the gut microbiota from damage due to antibiotics.
However, we also agree that the currently suggested oral antibiotics (non-absorbable + absorbable) must be evaluated in terms of SSI reduction as well as microbiota changes.
We have added a few sentences regarding oral antibiotic preparation and its effect on changes in the gut microbiota in the Discussion section of the revised manuscript (lines 357-365).
There might be some confounding factors that are not addressed in the paper and that should be specified. Potential risk factors for Anterior Resection syndrome are: level of anastomosis (low anastomosis have higher risk), total as opposed to partial mesorectal excision, obstructive presenting symptoms. As a matter of fact you don’t consider obstructive presenting symptoms as an exclusion criteria: does it mean no one among your patients had colonic obstruction? Please address all cofounding factors.
Table 1: p-value for comorbidities is not specified. Please address.
(Answer) Thank you for your comments.
We agree with your concern about the influence of various risk factors on the anterior resection syndrome (ARS). We might have been unable to control all surgical and patient factors that could influence the postoperative bowel function. However, we used strict inclusion criteria to exclude the operative factors associated with ARS. We excluded patients with endoscopic or symptomatic obstruction. Although the obstruction was relieved using stoma or stent insertion, patients with preoperative obstruction were excluded. We have clearly indicated this in the Methods section of the revised manuscript. (line 83-84)
The level of anastomosis was associated with postoperative defecatory dysfunction, as the reviewer has pointed out. In addition, anastomosis-related operative complications were significantly higher after rectal cancer surgery than after colon cancer surgery. Therefore, we included only patients with sigmoid colon cancer in this study. For sigmoid colon cancer, the level of anastomosis had less influence on the complications and dysfunction of the anastomosis. The ARS was more distinct after rectal cancer surgery, although bowel dysfunction was also reported after anterior resection of sigmoid for colon cancer or diverticultitis < Levack MM, et al. Sigmoidectomy syndrome? Patients' perspectives on the functional outcomes following surgery for diverticulitis. Dis. Colon Rectum 2012;55:10–17. ; Elfeki H, et al. Bowel dysfunction after sigmoid resection for cancer and its impact on quality of life. Br J Surg. 2019; 106:142–151.; van Heinsbergen M, et al. Bowel dysfunction after sigmoid resection underestimated: Multicentre study on quality of life after surgery for carcinoma of the rectum and sigmoid. Eur. J. Surg. Oncol. 2018;44:1261–1267.> Although strict inclusion criteria were necessary to set stable circumstances for the study, we thought that the differences in ARS might not be prominent in the present study because we included only those patients with sigmoid colon cancer who underwent anterior resection.
In the manuscript, description such as “adenocarcinoma of sigmoid colon and rectum” in the Methods section and Figure 1 would be confusing. Hence, in the Methods section, we clearly defined that we included patients with sigmoid colon cancer (line 81) and in Figure 1. For these patients, the level of anastomosis was not checked during the operation.
The p value of the comorbidities (p=0.227) has been added in Table 1.
The adverse events and complication paragraph should be rephrased. As it is not summarized in a table it is difficult to follow in the text. Line 189 you start with “of AEs occurring in 11 patients…” and in the following sentence “of 22 AEs occurring in the placebo group…”. So, you can’t compare how many patients with adverse events unless you mean that 11 patients in the probiotic group had adverse events and 22 patients in the placebo group. The p-value mentioned of 0.297 refers to the difference between AEs in the placebo vs probiotic group? 22 vs 11? No significance?
Which are the severe adverse events in the placebo group? And how do you explain them?
As far as complications is concerned, I would specify the infections developed in the two groups as there is concern that, in immunocompromised patients, probiotics can favor septic infections.
In the discussion you mention that adverse events of probiotic administration and placebo did not differ between the two groups, and all adverse events completely resolved by discontinuing the probiotic powder (Line 287-290). In the results chapter you never mention probiotic/placebo discontinuation. If so many patients discontinued the probiotic I don't think this drop out rate can support the conclusions you make.
(Answer) Thank you for your comments. AEs are important in studies on food or drugs.
AEs developed in 11 and 16 patients in the probiotics and placebo groups, respectively. Some patients in the placebo group experienced more than 1 AEs; therefore, 22 AEs were reported. The rates of AEs were 45.7% and 33.3% in the placebo and probiotics groups, respectively, and it was not statistically different (p=0.297). Severe AEs, such as anastomotic leakage and intractable diarrhea, developed in two patients in the placebo group, they were excluded. Among the reported AEs, those with clear association with the test powder were reported. However, we could not exclude the relationship of the two AEs with the test powder in the probiotics group (cholelithiasis and tremor) and three AEs in the placebo group (diarrhea, tremor, and abnormal liver function). Therefore, the rate of AEs with potential association with the test powder was not high in both groups (6.06% in probiotics vs. 8.57% in placebo) and not significantly higher in the placebo group.
Severe AEs in the placebo group were anastomotic leakage and intractable diarrhea, and these two patients were excluded. AEs that resolved while discontinuing the test powder were the above-mentioned AEs that could not be excluded such as cholelithiasis, abnormal liver function test, tremor, and diarrhea. These were resolved without any specific management after discontinuation of the test powder at the end of the study. As the reviewers have pointed out, the description in the discussion section was confusing “In this study, the incidence rates of AEs due to test and placebo powders did not significantly differ, and all AEs were completely resolved by discontinuing the probiotic powder.” It could be misunderstood that test powder ingestion was discontinued in all patients with AEs.
Therefore, we have revised this portion to improve clarity. In addition, we have revised the paragraph on AEs (line 192-197) and have added the supplementary table (Table S1) for detailed information in the results section and have added a few lines regarding AEs in lines 303-309 in accordance with the reviewer`s comments.
Septicemia related with probiotics has been reported in children and adults with immunocompromised conditions or in those with medical co-morbidities < Zein EF. Lactobacillus rhamnosus septicemia in a diabetic patient associated with probiotic use: a case report Ann Biol Clin (Paris). 2008;66:195-8. ; Kochan P, et al. Lactobacillus rhamnosus administration causes sepsis in a cardiosurgical patient--is the time right to revise probiotic safety guidelines? Clin Microbiol Infect 2011;17:1589-92.; Ee Groote MA, et al. Lactobacillus rhamnosus GG bacteremia associated with probiotic use in a child with short gut syndrome. Pediatr Infect Dis J. 2005 Mar;24(3):278-80.; Vahabenezhand E, et al. Lactobacillus bacteremia associated with probiotic use in a pediatric patient with ulcerative colitis. J Clin Gastroenterol 2013;47:437–439>. Although we excluded patients with severe medical co-morbidities, immunologic disease, and immunocompromised conditions to prevent septic complications, infectious complications occurred in both groups (2 in the probiotic group and 5 in the placebo group). In the probiotic group, cholecystitis and postoperative pneumonia developed, whereas in the placebo group, 2, 1, 1, and 1 patient developed fever, enteritis, cystitis, and, pseudomembranous colitis, respectively. However, there was no serious septicemia. We have added detailed information of the infectious complications in the Results section of the revised manuscript (lines 201-204).
In the discussion you should address the fact that the placebo group had more postoperative complications and more adverse events and possibly address it.
The study is well conducted and results are interesting but I do think that we should address all the confounding factors.
(Answer) We addressed the AEs and complications in detail in the Discussion section as well as in the Results section (line 192-197, line 292-301, 303-309).
We are thankful to the reviewers’ insightful comments. We agree with reviewer`s opinion about addressing confounding factors in detail and with clarity. We have revised the manuscript to improve clarity of the manuscript. The revisions are highlighted in yellow.

Reviewer 3 Report
I appreciate the opportunity to review this interesting study on the effects of probiotics on bowel dysfuntion after Anterior Colorectal Cancer Resection
I appreciate the opportunity to review this interesting study on the effects of probiotics on bowel dysfunction after Anterior Colorectal Cancer Resection
This is a randomized, double-blind trial that analysed the effects of probiotics given for four weeks, starting at one week preoperatively on anterior resection syndrome (ARS), quality of life inflammatory markers, and microbiota composition in patients with adenocarcinoma of the sigmoid colon or rectum indicated for anterior resection. The authors found some alleviation of postoperative bowel dysfunction and changes in microbiota composition and inflammatory markers.
I enjoyed reading the manuscript. I commend the authors for many strengths of their work, including addressing an interesting and timely question, careful planning and design of the trial, the proper conduct of the trial, and well-performed analysis.
The subject is in the range of the journal, and the manuscript is of clinical relevance. It is well written, and data are appropriately presented.
Considering these strengths, though, as I read the manuscript, I found some areas in which I would have appreciated greater clarity.
Comments:
Abstract
There is no information in the abstract about which patients this study concerns. Of course, because there is information that the research concerns ARS, we can guess, but it should be clearly written.
The conclusion that probiotics alleviated postoperative bowel dysfunction is not entirely justified.
Methods
Did all the patients come from one center?
Who generated the random allocation sequence, who enrolled participants, and who assigned participants to interventions?
Who did prepare probiotic and the placebo powder?
Were there differences in consistency or taste between them?
Has anyone checked this?
Results
There are considerable differences between groups, both in terms of pathological data and ARS score at baseline. The conclusions drawn are, therefore, not entirely convincing.
Discussion and Conclusions
The authors' conclusions (and title) are not entirely justified. I think that they should draw more conservative conclusions. There were no marked differences between the groups. There was no difference in QoL and Nutritional Screening Index. Observed changes in microbiota are difficult to interpret and mostly not statistically significant. Also, there was really no marked difference in ARS scores considering differences at baseline.
Author Response
Reviewer 3
I appreciate the opportunity to review this interesting study on the effects of probiotics on bowel dysfuntion after Anterior Colorectal Cancer Resection
This is a randomized, double-blind trial that analysed the effects of probiotics given for four weeks, starting at one week preoperatively on anterior resection syndrome (ARS), quality of life inflammatory markers, and microbiota composition in patients with adenocarcinoma of the sigmoid colon or rectum indicated for anterior resection. The authors found some alleviation of postoperative bowel dysfunction and changes in microbiota composition and inflammatory markers.
I enjoyed reading the manuscript. I commend the authors for many strengths of their work, including addressing an interesting and timely question, careful planning and design of the trial, the proper conduct of the trial, and well-performed analysis.
The subject is in the range of the journal, and the manuscript is of clinical relevance. It is well written, and data are appropriately presented.
Considering these strengths, though, as I read the manuscript, I found some areas in which I would have appreciated greater clarity.
Comments:
Abstract
There is no information in the abstract about which patients this study concerns. Of course, because there is information that the research concerns ARS, we can guess, but it should be clearly written.
The conclusion that probiotics alleviated postoperative bowel dysfunction is not entirely justified.
(Answer) We appreciate the comment. We have added details regarding the patients who were enrolled in this study (line 34-35).
In the conclusion section, we agree with your concern. We also think that we need to be careful while drawing conclusions. We have revised the Conclusion section to show the results of this study more clearly (line 46-47).
Methods
Did all the patients come from one center?
Who generated the random allocation sequence, who enrolled participants, and who assigned participants to interventions?
Who did prepare probiotic and the placebo powder?
Were there differences in consistency or taste between them?
Has anyone checked this?
(Answer) Four centers in Korea participated in this study; this has been described in the Methods section (line 73-74). Patients were enrolled from all centers. Patients who met the inclusion criteria were reported centrally to the clinical research agent, and screening was done. Patients who met the inclusion criteria and agreed to participate in the study were enrolled by the central clinical research agent. The randomization sequence was generated using a randomization number. The randomization table is a sequential application of random numbers (random numbers A and B) generated by the randomization program of the SAS® system, starting from clinical trial subject no. 1. This system is designed and reproduced using SAS®.
The probiotic and placebo powder were identical in appearance and taste and were sourced from the same manufacturer. They were checked by the clinical research agent.
The study process, including enrollment, randomization, test powder preparation, administration, and monitoring was centrally coordinated by a clinical research agent from NeoNutra Co., Ltd. (Seoul, South Korea). We have described the study process in further detail in the Methods section of the revised manuscript.
Results
There are considerable differences between groups, both in terms of pathological data and ARS score at baseline. The conclusions drawn are, therefore, not entirely convincing.
(Answer) We understand the reviewer’s concern. The two groups were not different in terms of demographic and pathologic data. However, the ARS score at baseline was different (Figure 2); the proportion of major and minor ARS was not statistically different between the two groups at baseline.
We, therefore, analyzed longitudinal changes within the groups to avoid comparison between the groups. As we show in Figure 2, the change in the ARS score from baseline to Visit 6 was less steep in the probiotics group (p=0.0527).
We also evaluated the change in each item in the ARS questionnaire (Table S2 in revised version). Among the items in the ARS questionnaire, flatus control showed significant improvement at Visit 7 from baseline (0.0331). The other items did not show significant differences from baseline.
As pointed out by the reviewer, it is not suitable to describe that probiotics “alleviate” postoperative bowel dysfunction. We also agree that we need to revise the conclusion based on the result. Therefore, we have revised the conclusion as follows:
These changes in the microbiota and inflammatory responses are owing to the use of probiotics before and after surgery, which reduce the postoperative bowel discomfort, as expressed by the ARS score.
Discussion and Conclusions
The authors' conclusions (and title) are not entirely justified. I think that they should draw more conservative conclusions. There were no marked differences between the groups. There was no difference in QoL and Nutritional Screening Index. Observed changes in microbiota are difficult to interpret and mostly not statistically significant. Also, there was really no marked difference in ARS scores considering differences at baseline.
(Answer) We appreciate your thoughtful comments.
We have revised the title as follows:
“Effects of PrObiotics on the symptoms and Surgical ouTComes after Anterior REsection of colon cancer (POSTCARE): a randomized, double-blind, placebo-controlled trial”
We have also revised the conclusion to be more conservative. The ARS scores did not show prominent changes from baseline, except for flatus control. However, changes in the ARS score would be less steep in the probiotics group, especially during the use of probiotics (Visit 6) (Figure 2). Although compositional changes in the microbiota are complicated and difficult to interpret, the result show more increase in the bacteria with fermentation activity in the probiotics group. Therefore, we drew the conclusion based on the result and have avoided drawing exaggerated conclusions.
